# Scanning Kelvin Probe for Detection in Steel of Locations Enriched by Hydrogen and Prone to Cracking

**Andrei Nazarov \*** **, Varvara Helbert and Flavien Vucko**

French Corrosion Institute, 220 Rue Pierre Rivoalon, 29200 Brest, France
* Correspondence: andrei.nazarov@institut-corrosion.fr; Tel.: +33-2980-515-52

**Abstract:** Hydrogen, due to corrosion processes, can degrade high strength steels (HSS) through embrittlement and stress corrosion cracking mechanisms. Scanning Kelvin probe (SKP) mapping of surface potential was applied, to visualize the locations with an increased subsurface concentration of hydrogen in mild steel and martensitic HSS. This work can help to determine the reasons behind hydrogen localization in a steel microstructure, leading to embrittlement and hydrogen-assisted cracking. Cathodic charging was used to insert hydrogen, which decreased the steel potential. Hydrogen effusion in air passivates steel, increasing the potential of HSS and mild steel. The passivation of steels was monitored depending on different conditions of cathodic pre-charging and the amount of absorbed hydrogen. The SKP could determine the area of diffusible hydrogen and the area of cracks. In addition, low potential locations linked to the hydrogen trapped in the deformed HSS microstructure were also determined, which delayed the steel passivation. Mild steel showed a uniform potential distribution related to interstitial hydrogen, without potential extremes attributed to locally accumulated hydrogen. Thus, SKP sensing can detect locations containing increased concentrations of hydrogen and sensitive to steel cracking.

**Keywords:** scanning kelvin probe; high-strength steel; embrittlement; cracking; passivation; hydrogen

## 1. Introduction

Hydrogen degrades the mechanical properties of many materials. Hydrogen-assisted cracking limits the application of high-strength steels (HSS), because their sensitivity to cracking increases proportionally with their strength [1–8]. Corrosion of steel or its metallic protective coating is the main source of hydrogen under service conditions. Galvanic coupling of HSS with sacrificial coatings, such as Zn-based coatings, facilitates hydrogen reduction and entry into the steel, which promotes hydrogen embrittlement in locations with external or residual stress. On the other hand, local anodic steel dissolution (e.g., in pits, cracks, crevices) due to hydrolysis of corrosion products leads to local acidification, which accelerates hydrogen reduction and diffusion in steel. Hydrogen accumulates in crack tips, interacts with the metallic bonds, and facilitates hydrogen-assisted crack growth [6]. These processes occur in different corrosive environments, including atmospheric weathering conditions [2]. In a thin layer of electrolyte, the distribution of ions and acidity is very non-uniform, which localizes hydrogen entry to the steel. Normally the anodic locations are more acidic and HSS has a more negative potential. These factors create favorable conditions for local hydrogen absorption during the atmospheric corrosion of steel [8,9].

To induce embrittlement, a critical diffusible hydrogen content must be reached [7,8,10]. This hydrogen concentration increases with deformation and correspondingly the concentration of dislocations. The locations prone to cracking contain an elevated stress field and a high subsurface hydrogen concentration [9]. The resistance of an alloy is strongly affected by the interaction of hydrogen with the microstructural heterogeneities, which act as hydrogen traps [11–14].

To determine the distribution of hydrogen and stress close to the steel surface, advanced electrochemical microscopy techniques must be employed [15–22]. It has been suggested that SKP and Scanning Kelvin probe force microscopy (SKPFM) are the most sensitive techniques currently available to detect locations with extremely low quantities of hydrogen [16,19]. It was determined that hydrogen can enrich stressed locations such as crack tips [14] and the grain boundaries [19]. SKP microscopy of steels was used to detect the locations of surface defects, built-in stress, and hydrogen [21–23]. Hydrogen and stress distribution in microstructures were indirectly characterized using SKP, measuring the potential drop in the surface oxide [23]. The links between the electron work function, oxide condition, and subsurface accumulation of hydrogen and stress are key points in hydrogen embrittlement. It was shown that surface re-oxidation was delayed as a function of the current density and duration of cathodic hydrogen pre-charging. Thus, the potential evolution during exposure to air can characterize the relative amount of subsurface hydrogen [23]. SKP mapping of a martensitic microstructure with locally developed residual stress and accumulated hydrogen displayed the lowest potential. It was shown that locations with developed tensile stress and a dislocation field attract hydrogen. Hydrogen pre-charged and pre-strained locations showed prolonged electron reactivity [23].

This work continued the studies in [23], and the objective was to apply SKP for mapping of the locations with an increased subsurface hydrogen concentration. A non-uniformity of the potential distribution was found in the HSS microstructure, which could be related to hydrogen enrichment of the stressed locations (hard spots) containing the pre-cracks, as well as any other microstructural discontinues, such as voids or second phase particles. The hydrogen localization in mild steel and HSS microstructures was compared.

## 2. Materials and Methods

### 2.1. Materials

The chemical compositions of mild carbon steel (SAE 1008) and high strength steel (HSS 1500) in this study are shown in Table 1, and the typical mechanical properties are presented in Table 2. The HSS steel microstructure was composed of martensite laths within prior austenite grains of 10–15 μm [23]. The SAE 1008 steel was fully ferritic and tested in annealed conditions.

**Table 1.** Chemical compositions of the investigated materials (wt. %).

|  | C | Si | Mn | Al | Nb | Cr | Mo | Ni |
|---|---|---|---|---|---|---|---|---|
| HSS 1500 | <0.3 | 0.18 +/−0.02 | 0.5 +/−0.1 | <0.1 | <0.1 | - | - | - |
| SAE 1008 | <0.13 | <0.10 | <0.5 | - | <0.08 | <0.15 | <0.06 | <0.2 |

**Table 2.** Mechanical properties of HSS 1500 and SAE 1008 steels.

| Material | Yield Strength | Tensile Strength | Elongation |
|---|---|---|---|
|  | (MPa) | (MPa) | (min %) |
| HSS 1500 | 1220–1520 | 1500–1750 | 3 |
| SAE 1008 | 350–550 | 650–880 | 8–25 |

Two sides of the steel samples were polished using 600–4000 grit SiC paper, degreased in acetone using an ultrasonic bath, rinsed in de-ionized water, and dried using compressed air that was free from oil.

### 2.2. Electrochemical Treatment and Measurements in Aqueous Electrolytes

The cathodic charging of steels was carried out either in 0.1 M NaOH or in 0.3 M $Na_2SO_4$ acidified to pH 2 through addition of $H_2SO_4$ aqueous electrolytes using a three-

electrode cell containing a working electrode (plate 20 × 20 × 0.5 mm), a counter Pt-mesh electrode, and reference saturated Hg/HgO or Hg/HgCl$_2$ electrodes. A "Gamry Reference 600" potentiostat was used for galvanostatic cathodic polarization at different cathodic current densities and durations. After treatment, the specimen was quickly rinsed in de-ionized water, dried in a stream of dry air, and installed in the SKP chamber for measurement. To perform local steel hydrogenation, an electrochemical cell was attached to the surface using an O-ring. Thus, a hydrogen-enriched area with a diameter of 9–10 mm was produced.

The open circuit potential (OCP) of the steel samples as a function of the prelaminar cathodic treatment (current density −5 mA/cm$^2$) was measured in 0.1 M NaOH aqueous electrolyte using a standard Hg/HgO electrode. The cycle consisted of cathodic polarization for 133 min, followed by OCP monitoring for 120 min. After 15 cycles, the sample was investigated using SKP. All experiments were carried out at room temperature (21+/−2 °C).

*2.3. Scanning Kelvin Probe*

In this study, a height-controlled SKP instrument from Wicinski & Wicinski GbR (Erkrath, Germany) was used. The reference electrode was a CrNi alloy needle with a tip diameter of about 100 μm, and the distance to the surface of working electrode was approximately 50 μm. Surface contour mapping (topographic profile) was performed simultaneously with the potential mapping. The measurements were carried out at an ambient air at humidity close to 60% RH and room temperature (21+/−2 °C). Prior to the measurement, the potential of the probe was calibrated above a saturated Cu/CuSO$_4$ electrode, and all potentials are provided versus the standard hydrogen electrode (SHE). To verify the stability of the potential of the SKP tip during long time measurements, calibration was additionally performed after the experiment.

The principle of SKP hydrogen detection is based on the interaction of hydrogen with the surface oxide film [19]. Thus, SKP sensing of hydrogen must be discussed in detail. SKP measures the contact potential difference between two metallic electrodes (the working and the probe) separated by an air gap. The potential of the SKP tip is calibrated relative to the reversible electrode, which makes it possible to determine the potential of the working electrode relative to a reference. The main factor contributing to the potential of the working electrode is the potential drop ($X_w$) across the metal/air interfaces (Equation (1)) [24,25].

$$X_w = \frac{\mu_{ox} - \mu_e}{e} + F_b + \beta_{ox/air} \tag{1}$$

The drop $X_w$ is mainly concentrated in the surface oxide film (Equation (1)), where the main contribution is from the contact potential difference at the metal/oxide interface, which is proportional to the difference in the chemical potentials of the electron in the metal ($\mu_e$) and in the oxide ($\mu_{ox}$) [25]. $F_b$ is the potential drop in the semiconducting oxide (near-surface bands bending) due to interactions with the environmental components. The $\beta_{ox/air}$ is the drop of the potential at the double electric layer oxide/electrolyte or oxide/air interfaces.

The objective of a number of works has been developing a protocol for SKP as a technique for the local quantification and distribution of the subsurface hydrogen, as a function of the microstructure, hydrogen traps, and stress [16,19–23]. This quantification is connected to the ability of hydrogen to chemically reduce the oxide film, which decreases the potential drop in the film (Equations (1)–(3)). It is important to note that, after calibration of the SKP tip above standard reversible red-ox electrode, the technique measures the electrochemical potential of the steel (working electrode) on a convenient electrochemical scale [26]. This makes it possible to compare potentials measured in air using the SKP and in the electrolytes using a reversible reference electrode.

The potential drop steel/air $X_w$ depends on the oxide film thickness and composition. The influence of oxide thickness on the potential was discussed in works related to the passivity [27,28]. A linear dependence between the potential and the oxide film thickness was

shown for pure metals and Fe-Ni alloys [27]. Equation (2) shows the connection between the overall potential drop metal–electrolyte of a passive metal ($\varphi_M$), the corresponding oxide film thickness ($L$), and the oxide film dielectric constant ($\varepsilon_{ox}$):

$$\varphi_M = \varphi_H \left( 1 + \frac{L}{\partial} \frac{\varepsilon_H}{\varepsilon_{ox}} \right) \tag{2}$$

where $\varphi_H$ is the potential drop across the Helmholtz layer of the electrolyte, and $\partial$ and $\varepsilon_H$ are correspondingly the thickness and dielectric constant of the Helmholtz layer.

For the metal oxides in different oxidation states and forming the red-ox couple, the potential drop in oxide phase follows the Nernst equation:

$$\mu_{ox}/e = const. + RT/F \ln \left( a_{Fe}{}^{3+}/a_{Fe}{}^{2+} \right) \tag{3}$$

where $a_{Fe}{}^{3+}$ and $a_{Fe}{}^{2+}$ are the activities of the iron species, $T$ is the temperature, and $R$ and $F$ are fundamental constants. This relationship between the degree of oxidation of the mixed oxides in the layer covering the iron was proven by potential measurements using SKP [29,30]. Thus, we can suppose that the hydrogen in the steel has an influence on the potential, due to the change in the oxide film thickness (Equation (2)) and the ratio of activities of iron species (Equation (3)). These processes can potentially determine SKP measurements. However, modelling of the potential as a function of the ratio of iron species activity in the film (Equation (3)) can be too simplistic. The potential of the surface film ($\mu_{ox}$, Equations (1) and (3)) is determined using flat band potentials, band gaps, and band bending ($F_b$, Equation (1)) for the number of oxides and hydroxides with different degrees of oxidation [29]. Thus, more work is needed to correlate the potential measured using SKP with film composition and thickness.

### 2.4. Scanning Electron Microscopy

A Hitachi SU3500 scanning electron microscope (SEM) connected to a Thermo-Scientific Ultra dry NSS 312 SEM/EDX energy dispersive X-ray spectrometer (EDX) was used to evaluate the surface morphology of the steel surface as a function of the surface treatment and to detect the presence of cracks.

## 3. Results and Discussion

### 3.1. SKP Monitoring of Steel Surface in Air after the Grinding and Hydrogen Charging

The primary objective was to evaluate the influence of oxide film formation/reduction on the steel potential. The native oxide film was removed by gentle mechanical grinding, the surface was degreased using ethanol and the SKP potential monitoring began about 5 min after grinding in ambient atmospheric conditions. The potential increase during exposure in ambient air followed the power law of the value close to the initial (Figure 1). In semi-logarithmic coordinates, the plot of the potential vs. time was a straight line (Figure 2). Thus, the formation of new oxide film increased the potential from 80 mV to 270–300 mV (SHE), and the potential could be considered linked to the oxide film thickness and dielectric constant (Equation (2)). Mild steel passivated slightly faster relative to the HSS. Accordingly to XPS data [30], the oxide films formed on iron surfaces in humid atmosphere consist of an interlayer $Fe_3O_4$ and a top layer of $FeOOH$, and the limiting step of growth is the migration of ionic vacancies through the film. The thicknesses of the oxide films formed in humid and dry atmospheres were 2 nm and 3.5 nm, respectively [31].

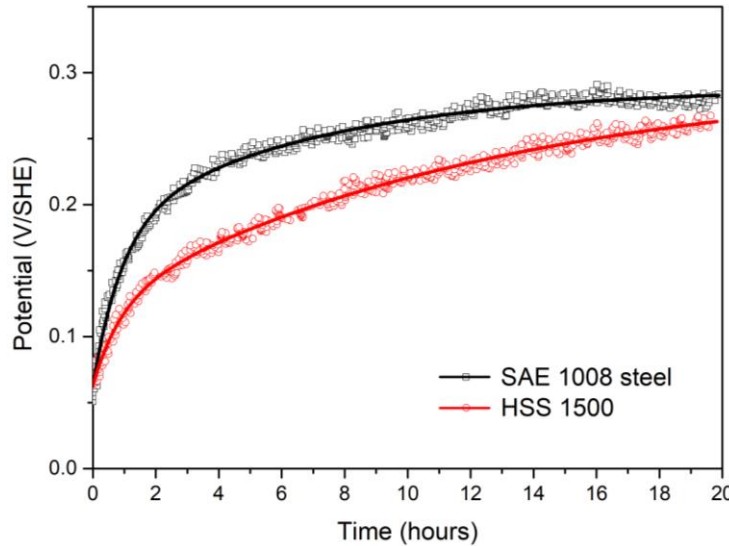

**Figure 1.** Monitoring of the potential of the steels in air at 60% RH and 22 °C after grinding.

**Figure 2.** Potential monitoring of the steel after surface grinding (mild steel ground and HSS 1500 ground) and after grinding followed by cathodic treatment in 0.1M NaOH aqueous electrolyte (HSS 1500 CP and mild steel CP) with a current density of $-5$ mA/cm$^2$ for 2.22 h (**A**), $-5$ mA/cm$^2$ for 4.44 h (**B**) and $-25$ mA/cm$^2$ for 24 h (**C**).

The SKP assessment of mild steel SAE 1008 and HSS 1500 was carried out after grinding and followed cathodic hydrogen charging in a 0.1 M NaOH aqueous electrolyte. Cathodic polarization applied a current density of $-5$ mA/cm$^2$ for 133 min, followed by surface rinsing in de-ionized water and quick drying in a stream of dry air. SKP potential monitoring was carried out in ambient air (22 °C and 60% RH) immediately after the surface treatment. The variations in the potential after grinding and after grinding–cathodic polarization are presented in Figure 2 as semi-logarithmic coordinates. The potential of the hydrogen charged surfaces was lower compared with the potential of the recently ground surfaces. The potential after grinding reached an initial value close to 0.3 V/SHE. On the other hand, after cathodic charging and prolonged exposure in air, the surface obtained a lower potential, in the range 170–180 mV (Figure 2A). Thus, hydrogen in steel can inhibit the surface oxide formation for a longer time compared with a mechanical removal, in line with [23]. The two steels showed a similar effect of the absorbed hydrogen on the kinetics of film formation. To increase the hydrogen concentration in the steels, the duration of the cathodic polarization was doubled (4.44 h). The SKP potential was monitored in air (Figure 2B). The initial potentials of the steels $-0.2$ and $-0.25$ V (SHE) were more negative than those of samples that underwent polarization for 2.22 h (Figure 2A), which could be attributed to an increased subsurface hydrogen concentration. However, after initially low values, the potential increased faster in a relatively shorter charging time (2.22 h) (Figure 2). This could have been the result of increased steel surface reactivity relative to the oxide formation. The potential monitoring after prolonged hydrogen charging with an increased current density is displayed in Figure 2C. The potential after charging was $-0.3$–$-0.35$ V (SHE), which was the most negative potential obtained after cathodic polarization in this study, and slightly higher than the thermodynamic value of the oxide formation Fe$^0$/Fe$^{2+}$ ($-0.44$V, SHE, 25 °C [32]) electrode. It is likely that the surface was close to an oxide-free condition, which is in line with [33]. The steels oxidation was significantly delayed (especially in the mild steel) relative to similar measurements. This could have been the result of the increased subsurface concentration of hydrogen and prolonged inhibition of oxide formation.

### 3.2. Effect of Cathodic Hydrogen Charging on the Rest Potential of HSS Steel in Aqueous Electrolyte

To date, studies of the effect of hydrogenation on surface reactivity and oxidation have been very limited. The kinetics of steel passivation after hydrogen charging were measured in an alkalic aqueous electrolyte. The hydrogen was inserted using cathodic polarization (duration 2.22h, current density $-5$ mA/cm$^2$), followed by the monitoring of the rest potential (duration 2h) using an Hg/HgO electrode in 0.1 M NaOH aqueous electrolyte. Figure 3A shows the variation of the rest potential of HSS 1500 as a function of the number of cycles. The cathodic polarization shifted the potential to negative values and then, during resting, the potential increased due to surface oxidation. A power law dependence was observed that correlated with the SKP measurements (Figures 1 and 2). Only the first cycle of the cathodic polarization of HSS (Figure 3A) showed the lowest potentials. Increasing the number of cathodic cycles increased the potential to more positive values, which was likely related to the formation of a thicker oxide–hydroxide film. After six cycles, the rest potential became more positive relative to the open circuit potential measured prior to cathodic polarization (Figure 3A). The passivation process in alkalic electrolyte was studied for a mild steel surface. Figure 3B summarizes the rest potential at 2 h as a function of the number of cycles for HSS and mild steel. HSS increased the rest potential more quickly relative to the mild steel, which may have been related to more effective formation of the surface phases in the NaOH aqueous electrolyte. This result is in line with passivation in air (Figure 2), which could have been the result of different efficiencies of hydrogen diffusion/effusion from the microstructure.

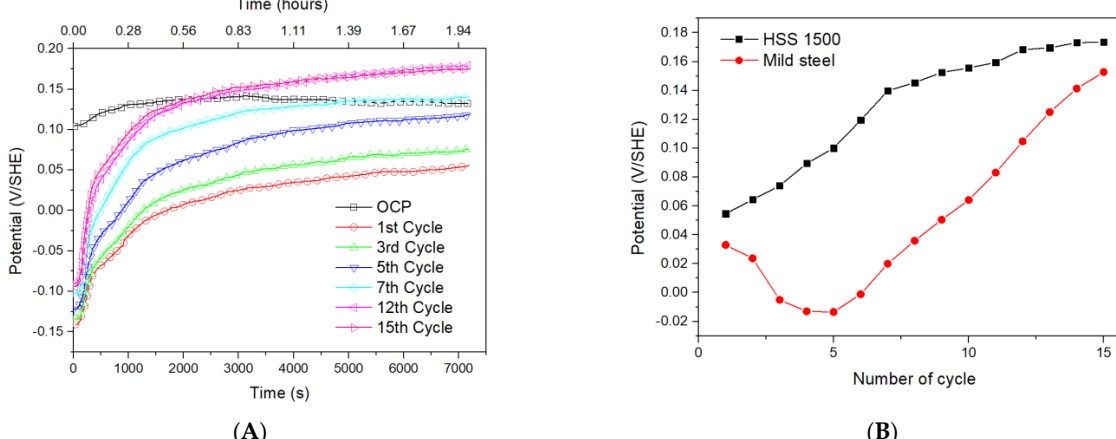

**(A)**                                                                 **(B)**

**Figure 3.** (**A**) Monitoring of the rest potential of HSS 1500 using Hg/HgO saturated reference electrode in 0.1 M NaOH aqueous electrolyte, after cathodic treatment with a current density $-5$ mA/cm$^2$ for 2.22 h. (**B**) the rest potential measured at 2 h for HSS 1500 and mild steel vs. the number of cathodic treatment cycle.

The HSS samples after cycling in aqueous electrolyte were rinsed in de-ionized water, dried, and the potential was monitored using SKP. The potential variations in air after cathodic charging at $-5$ mA/cm$^2$ for different durations (1 cycle 2.22 h) are compared in Figure 4. Cathodic polarization treatment for 15 cycles in 0.1 M NaOH showed the most positive potential, which could have been related to the formation of a thicker surface film. More negative potentials were found for shorter times of surface hydrogenation, which enabled the formation a thinner oxide–hydroxide film. Thus, the hydrogen absorption and the oxide–hydroxide film formation influenced the potential in opposite directions. For SKP detection of hydrogen-enriched locations, steel oxidation and exposure of electrolytes must be avoided.

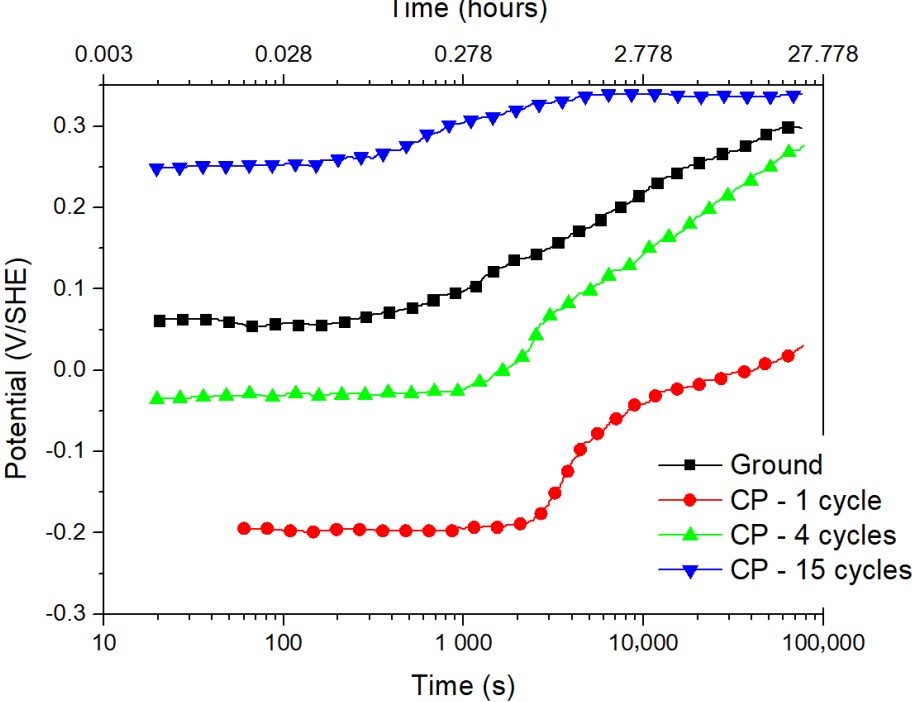

**Figure 4.** SKP monitoring of the potential of the above HSS 1500 steel in air after cathodic polarization in 0.1 M NaOH aqueous electrolyte with a current density of $-5$ mA/cm$^2$ for different numbers of cycles. The reference HSS surface was measured after grinding.

According to literature data [13], depending on the concentration and diffusivity of hydrogen in steel, the emission of diffusible hydrogen after cathodic polarization can continue for about 1 h. The atomic hydrogen is strongly reductant ($E^0_{H^0/H^+}$ = −2.106 V, SHE, 25 °C [32]) and can reduce different oxidizers, such as the iron oxide–hydroxide species and adsorbed oxygen (Equations (4) and (5)). The oxygen in air oxidizes the steel surface (Equations (7)–(10)), forming an oxide–hydroxide film that increases the potential [23,34,35]. All these reactions can have an influence on the rest potential of hydrogenated steel in air or aqueous electrolyte. However, the range of the measured potentials (Figures 2–4) corresponds to the reduction/oxidation of iron oxides and the coverage of the steel by these species (Equations (7)–(10)). Thus, these processes probably control the potential, which is in line with [35]. However, the contribution of oxygen reduction reactions (Equation (5)) to steel potential cannot be excluded. In this case, atomic hydrogen oxidation can take place on the surface of a relatively active electrode (Figure 2), before formation of a passive film that prevents the movement of electrons to oxygen molecules [29]. The catalytic properties of the surfaces, which depend on the steel composition and contaminations, can influence the efficiency of reactions (4)–(10). Thus, oxides of less noble elements can remain after hydrogen charging and promote catalytic effects. From this point of view, oxygen and hydrogen reduction reactions were investigated and discussed using a noble electrode, such as Pd [36].

$$H^0 + Fe^{3+} \rightarrow H^+ + Fe^{2+} \tag{4}$$

$$4H^0 + O_2 \rightarrow 2H_2O \tag{5}$$

$$H^0 + H^0 \rightarrow H_2 \tag{6}$$

$$Fe^0 + 2H_2O \rightarrow Fe(OH)_2 + 2H^+ + 2e \tag{7}$$

$$2Fe^{2+} + 6e + 3/2\,O_2 \rightarrow Fe_2O_3 \tag{8}$$

$$Fe^{2+} + 2e + 4Fe_2O_3 \rightarrow 3Fe_3O_4 \tag{9}$$

$$2Fe^{2+} + 1/2\,O_2 + 3H_2O \rightarrow 2FeOOH + 4H^+ \tag{10}$$

### 3.3. SKP Assessment of HSS Hydrogen-Assisted Cracking

An alkalic electrolyte was used, in order to decrease the corrosion of the steel after cathodic hydrogenation. However, to generate cracks in HSS, the sample was polarized under severe hydrogen charging conditions, using 0.3 M $Na_2SO_4$ acidified by sulfuric acid to pH 2, for 25 h with an applied current density of −25 mA/cm$^2$. After treatment, to inhibit corrosion, the surface was passivated in a 0.1 M NaOH aqueous electrolyte, rinsed in de-ionized water, and dried in a stream of compressed air. The sample was examined using a SKP (Figure 5) and SEM (Figure 6). Two areas with more positive and more negative potentials were found on the SKP map. SEM showed many one-directional oriented cracks related to low-potential areas (Figure 6). Thus, the cracking area was localized and corresponded to altered HSS microstructures, such as the presence of hard spots [4]. The deformation field of cracks attracted hydrogen from the alloy bulk [13,23,37–39]. A fine microstructure in the SKP map corresponding to a single crack was not found, due to lateral hydrogen diffusion. Thus, the potential distribution in the area of cracks was uniform at the level of the spatial resolution of the technique.

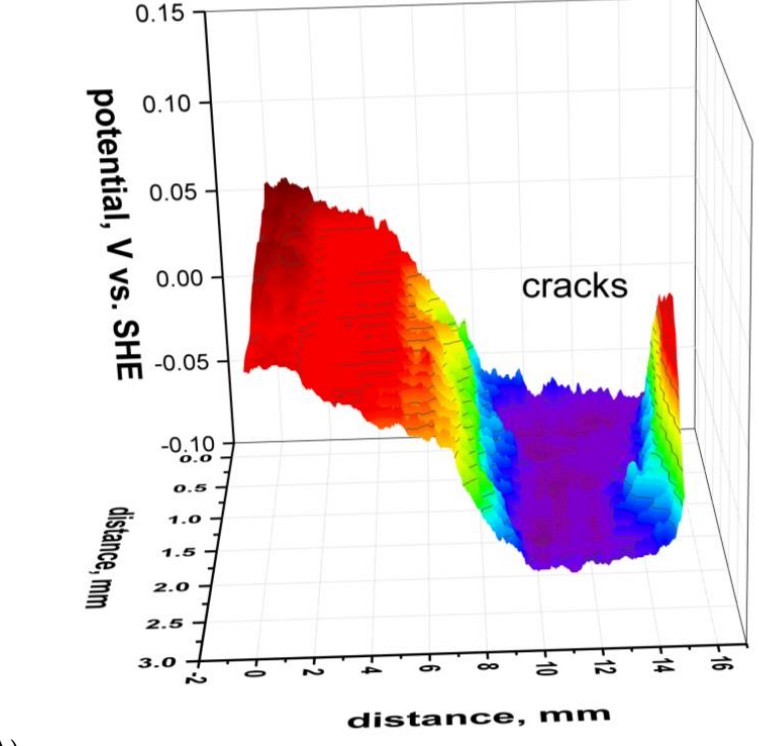

(**A**)

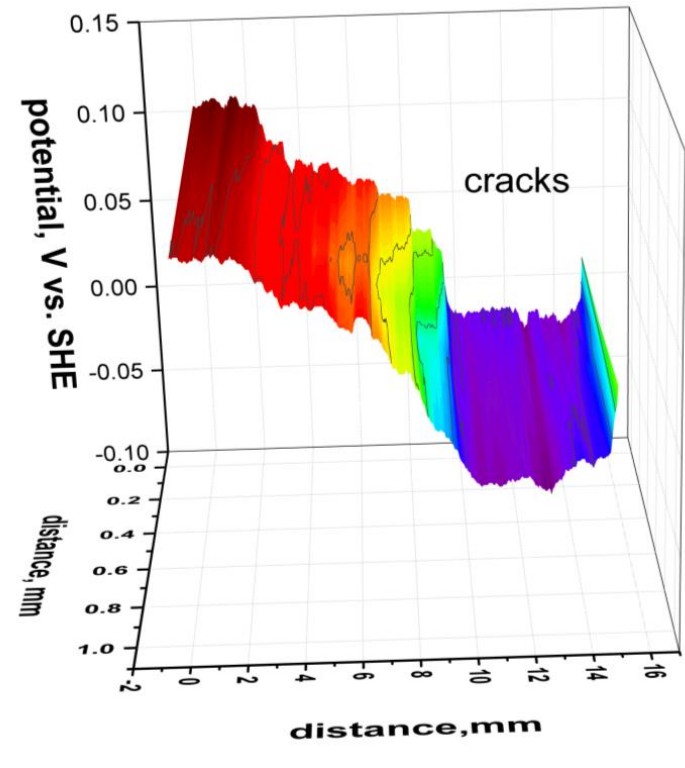

(**B**)

**Figure 5.** SKP potential distribution for HSS 1500 steel after cathodic polarization with a current density of −25 mA/cm² in 0.3 M Na₂SO₄ (pH 2) aqueous electrolyte for a duration of 25 h. The measurements were performed in air at 60% RH, after cathodic charging at 4 h (**A**) and 24 h (**B**).

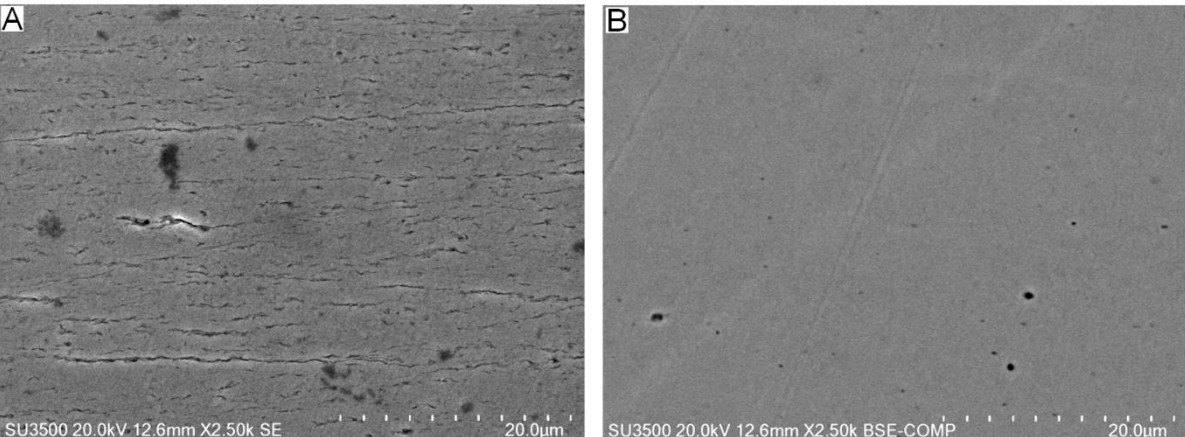

**Figure 6.** SEM images of the HSS surface after the cathodic treatment with a current density of $-25$ mA/cm$^2$ and a duration of 25 h in 0.3 M Na$_2$SO$_4$ at pH 2: (**A**)—the area with negative potentials, containing cracks; (**B**)—the area with positive potentials.

To characterize the area of the cracks in more detail, the potential above the surface was monitored during the exposure to ambient air (Figure 7). These experiments could be used to determine the rate of steel passivation. The area of the cracks had a lower rate of passivation during exposure to a standard atmosphere (22 °C, 60%RH). This dependence was compared with similar measurements for mild steel after cathodic pre-charging in the same electrolyte and using the same current density. The mild steel did not show any cracking, and the surface passivation proceeded similarly to the uncracked HSS area (Figure 7). Thus, it was supposed that the low potential area contained the cracks obtained due to the increased hydrogen concentration with a long activation of the HSS surface. In a previous study, SKP mapping of plastically deformed and hydrogenated HSS also showed relatively long passivation times for the pre-strained locations, due to the accumulation of hydrogen [23]. The experimental data can be explained by the Zakroczymski [40] model of hydrogen state in steel, showing that effusion from the traps takes significantly longer relative to the interstitial hydrogen.

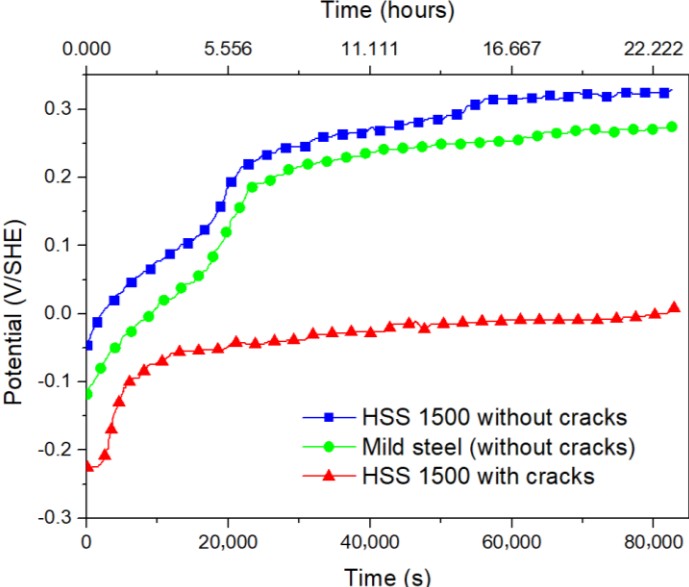

**Figure 7.** Monitoring of the potential of the mild steel and HSS 1500 in air after cathodic polarization at $-25$ mA/cm$^2$ during 25 h above the surface with the cracks in HSS 1500 and without cracks (HSS1500 and mild steel).

### 3.4. SKP Detection of Hydrogen in the Steel Membrane

The data showed that hydrogen absorption decreased the potentials (Equation (4)), but steel oxidation using an electrolyte (Equations (7)–(10)) increased the potential. This makes SKP evaluation of the hydrogenation of the directly pre-charged steel surface complicated. It was shown that SKP could detect the hydrogen flux using a permeation membrane setup [9,33–35]. In this case, the diffusible hydrogen interacts with the side of the membrane in contact with dry lab air. The mild steel and HSS membranes were locally pre-charged for 1.5 h using a 0.1M NaOH aqueous electrolyte and current density $-5 \, \text{mA/cm}^2$, then rinsed in deionized water and dried. SKP mapping (Figures 8 and 9) was applied to the direct and permeation side of the steel membrane (thickness 0.5 mm). Figure 8 shows that diffusible hydrogen uniformly decreased the potential of mild steel by 150–180 mV. The directly polarized side uniformly decreased the potential by 200–250 mV. These values show the partial steel passivation due to the 3 h exposure in air. The distributions are relatively uniform and correspond to the effusion of interstitual hydrogen from the steel bulk. A small decrease of the potential at the edge (Figure 8A) could have been an effect of lateral hydrogen diffusion from the charging area.

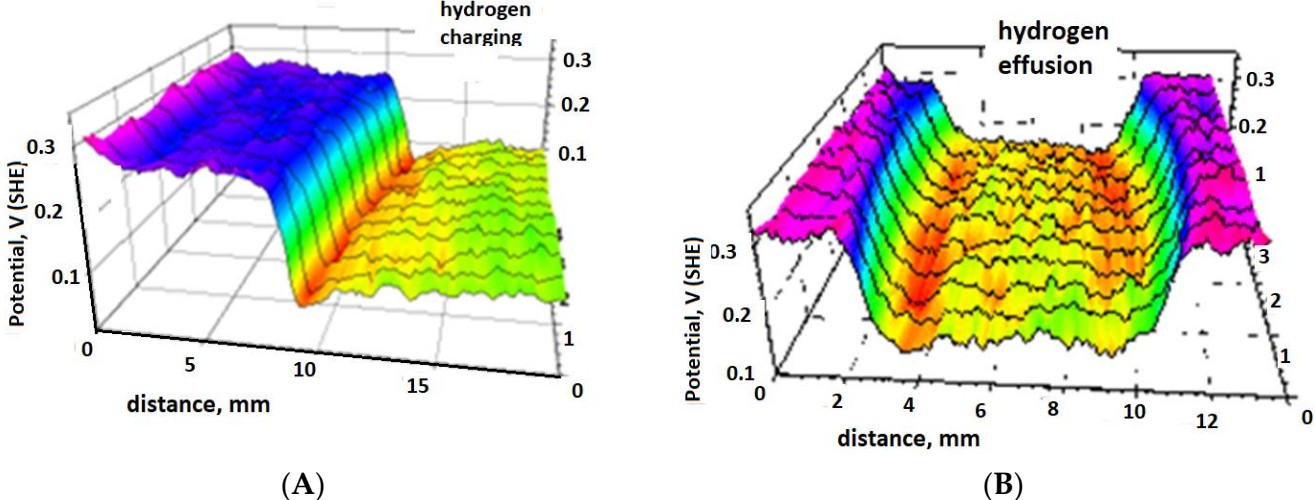

**(A)**　　　　　　　　　　　　　　　　　　　　**(B)**

**Figure 8.** Potential distribution of mild steel SAE 1008, directly polarized (**A**) and permeation (**B**) sides of the membrane. The steel was locally pre-polarized for 1.5 h in 0.1 M NaOH aqueous electrolyte using a current density of $-5 \, \text{mA/cm}^2$. SKP measurements were carried out in air at 60%RH 3 h after treatment.

For the HSS 1500 membrane, a similar surface treatment and SKP measurement was carried out (Figure 9). After 3 h of exposure in air, the directly polarized side showed a uniform decrease of potential from 0.26 V to 0.03–0.06 V (SHE). The map (Figure 9A) also contained low potential wells at $-0.18$ and $-0.25$ V (SHE), which were probably related to hydrogen localization. For example, local deformation fields or inclusions can accumulate hydrogen and develop blisters and microvoids [39].

The cathodic treatment uniformly decreased the potential of the permeation side of the membrane by 150 mV (Figure 9B). The scanning of the permeation side was repeated after 20 h of exposure in air (Figure 9C). The potentials increased, showing hydrogen effusion and surface oxidation due to contact with air. This potential non-uniformity could have been the result of hydrogen and surface oxide distribution.

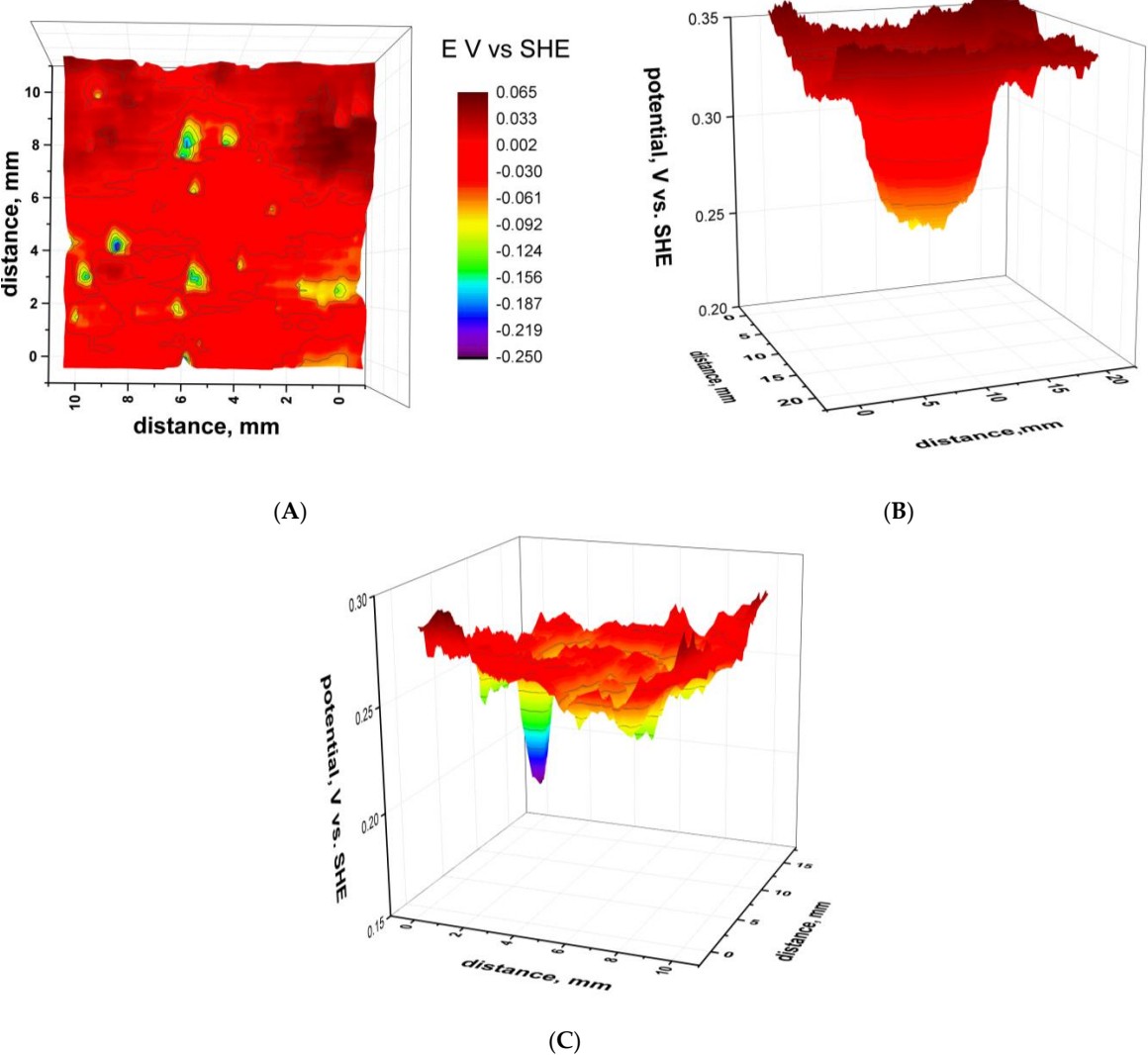

**Figure 9.** Potential distributions of the directly polarized (**A**) and permeation (**B**,**C**) side of HSS 1500 steel membrane (thickness 0.5 mm). The steel was charged using a current density of −5 mA/cm² in 0.1 M NaOH for 1.5 h. SKP measurements were carried out in air at 60%RH 3 h after treatment (**A**,**B**) and 20 h (**C**).

As was mentioned in [23], the influence of cathodic polarization on the steel potential of the permeated side was significantly lower compared with impact on the directly polarized side. On the directly polarized side, nearly all oxides were completely reduced, but the permeated hydrogen only partially reduced the oxides on the diffusion side of the membrane. Even high cathodic currents did not decrease the potential of the diffusion side to lower than 0.12 V (SHE) [23]. However, in some experiments with HSS, spikes and wells were also found in maps of the diffusion side of the membrane. For example, Figure 10 shows the circle area of the diffusible hydrogen and well at −0.25 V (SHE), which may be related to the accumulated hydrogen.

The hydrogen trapped in the microstructure of steel bulk can be visualized using SKP [21,23,41]. The HSS sample was pre-charged for 4.4 h at −5 mA/cm² in 0.1 M NaOH aqueous electrolyte, rinsed in deionized water, and exposed to dry air above silica gel in a desiccator for 10 days. Afterwards, the direct and permeation sides were gently ground using P4000 emery paper. To create a new oxide film, the specimen was exposed in dry air for 24 h. The potential distribution in the affected areas was uniform and no locations with extreme potential values was found. This could have been the effect of the lateral diffusion reaching the equilibrium of hydrogen fugacity in the material. Figure 11 shows

linear scans across the polarized (right hand side) and non-polarized (left hand side) parts of the membrane. It is obvious that, even with removal of the oxide film, the newly grown film showed a lower potential in the treated area, where the alloy likely contained residual hydrogen. The effect was more pronounced on the side that was directly polarized, rather than on the opposite side, which may have been the result of a larger subsurface hydrogen concentration being trapped in the microstructure. Thus, the SKP technique is sensitive to even very small amounts of stored hydrogen and even when used a long time after the hydrogen treatment occurred.

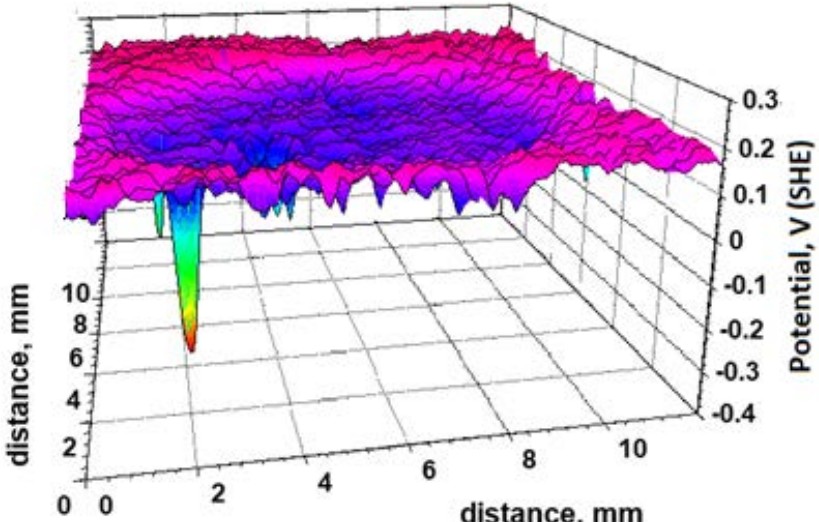

**Figure 10.** Potential map of the permeation side of the HSS 1500 steel membrane (thickness 0.5 mm). The steel was pre-charged using a current density of $-10$ mA/cm$^2$ in 0.1 M NaOH for 3 h. SKP measurements were carried out in air at 60%RH 1.5 h after treatment.

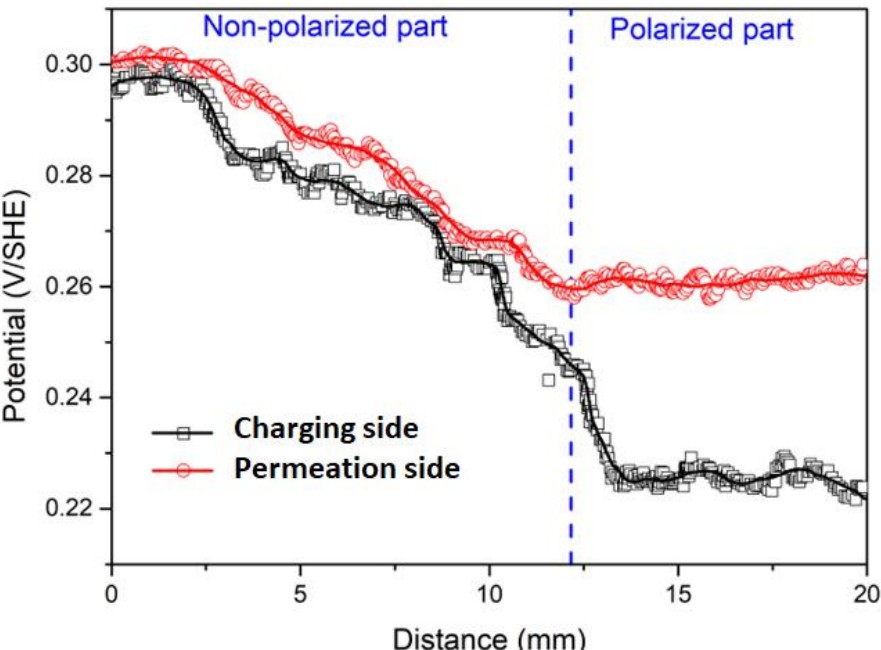

**Figure 11.** Linear scans across the directly polarized and permeation sides of an HSS 1500 membrane, after treatment of $-5$ mA/cm$^2$ in 1 M NaOH for a duration of 4.4 h. The sample was exposed to dry air for 10 days and then ground with emery paper. SKP linear scans were measured 24 h after grinding.

## 4. Conclusions

Application of the SKP technique for assessing the hydrogen distribution in mild and HSS steels was investigated. Indirect sensing was based on the measurement of the potential of the oxide-hydroxide layer, which was influenced by the hydrogen flow effused from the traps in the steel.

From the results, the following conclusions were drawn:

1.  Cathodic polarization and hydrogen absorption decreased the potential of the steels, due to the reduction of the surface oxide film and hydrogen absorption. Exposure to air or alkali electrolyte increased the potential, due to hydrogen effusion and the formation of oxide-hydroxide phases. The increase in the amount of hydrogen trapped in the microstructure delayed the steel oxidation and the noble shift of the potential.
2.  Cathodic hydrogen charging under severe conditions was used to induce cracks. SKP showed low potentials for this area, which was the result of hydrogen accumulation in the deformation field of cracks. The cracks showed the slowest kinetic of oxide formation (passivation) in air.
3.  The hydrogen permeation across the steel membrane was studied. The interstitial hydrogen uniformly decreased the potential of the entry and diffusion sides for mild steel and HSS. Additionally, negative wells and spikes of potential were found in the maps of HSS for the directly hydrogen charged and hydrogen permeation sides of the membrane. The locations enriched by hydrogen obtained a lower potential, due to the prolonged effusion and reduction activity.

**Author Contributions:** Conceptualization and methodology, A.N. and F.V.; Investigation and data curation, F.V. and V.H.; writing—original draft preparation and editing, A.N. and V.H.; Funding acquisition, F.V. All authors have read and agreed to the published version of the manuscript.

**Funding:** This project has received funding from the Research Fund for Coal and Steel under grant agreement No. 101034041.

**Data Availability Statement:** Upon request to the authors.

**Conflicts of Interest:** The authors declare no conflict of interest.

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
