# Peer review of "Scanning Kelvin Probe for Detection in Steel of Locations Enriched by Hydrogen and Prone to Cracking"

_cmd, doi:10.3390/cmd4010010_

Round 1

Reviewer 1 Report

This is certainly a work suitable for publication, but some issues need to be addressed.

 1.First of all, here Kelvin probe measurements have been performed on the passive layers of an HSS and a mild steel have been performed on the hydrogen charging side and the other side (referred to as permeation/diffusion side).

That the potential of the oxides changes due to the presence of sufficient activity of hydrogen is already well known. The main message seems to be here that the according effect changes with time after charging. This should maybe made clearer.

 Furthermore, the authors discuss the contributions that contribute to the potential of the surface oxides in eq. 1-3. Further they state:

“Thus, we can suppose that hydrogen in the steel can influence on the potential due to change the oxide film thickness (Eq. 2) and the ratio of activities of iron species (Eq. 3). “ and “On the other hand, it is possible that the steel electrode potential can be controlled by the reactions of hydrogen with the oxidizers containing in air (moisture and oxygen) [9,16, 20].”

 Certainly, one can be a bit more precise. And hence the according discussion can be improved.

 For this one has to consider that the potential of an oxide exposed to air is defined by its band gap, the Fermi level position within the band gap and band bending at the surface    (Hausbrand et al, J. Electrochem. Soc. 155(7) (2008) C369-C379 ). The band bending is due to the fact that on the dry (non immersed) oxide surface the anodic reaction is significantly suppressed, more than it is the case for oxygen reduction. Hence the surface charges positively, which causes the band bending, until no further tunneling of electrons to adsorbed oxygen is possible. The band bending required for that is typically between 50 and 100 mV (see Hausbrand et al). That comes on top of the flat band potential, which is determined by the band gap and the defect density (donor and acceptor states). Now depending on the degree or partial reduction of the oxide also the band gap will be affected (by the different kind of iron oxides possible and the percentage of hydroxide; hydroxides have a lower band gap (see Piazza et al., Electrochim. Acta 48 (2003) 1105; Di Quarto et al., Russ. J. Electrochem., 36 (2000) 1358). Hence, in the case we are considering here the oxides will change a lot during the measurement and changes of band gap cannot be ruled out, i.e. e.g. the ratio between Fe2+ and Fe3+ is certainly a too simple parameter (valid only within the same oxide: on the induced changes see e.g. Luo et al. Electrochem. Commun. 79 (2017) 28-32 ).

 This is especially true for the charging side. On the charging side the oxide is (during cathodic charging) most likely completely reduced (certainly the iron oxide part, for the HSS it depends on the oxide of the less noble elements what really remains during charging).  That should be better discussed.

This is already the first difference between the two sides: on the charging side a completely new oxide forms in the moment the charging is stopped, but under the influence of initially very high hydrogen activity. On the diffusion side the existing oxide is partially reduced by the hydrogen.

At the same time, as correctly pointed out by the authors, oxygen (from the air) will cause (re-)oxidation (on both sides).

So many factors come together and the effect of alloying elements might play a possible role in catalytic activity as well as in defining the defect density in the band gap.

Hence, in a comparison between the responses of mild steel and HSS has to be made with great care, although they behave quite similar (at least initially). Since the hydrogen permeation rate plays also a crucial role here, it seems that for both materials the permeation rates and also the response of the oxide sis quite similar, at least initially. The re-oxidation at later stages seems to be a bit different.

Concerning potential, it can be said is, that as long as the hydrogen permeation towards the surface is still relatively high, i.e. most likely in the first 1000 s where the lowest potentials are measured, the potentials are most certainly defined by the equilibrium between oxygen reduction and hydrogen oxidation, modified by the different catalytic properties of the surface. This is because due to high H permeation rates anodic H oxidation is possible as counter reaction, i.e. band bending will be negligible.

On noble metals such as Pd such a formation of ORR-HOR equilibrium was recently reported (see Zhong et al., Chemelectrochem 8(4) (2021) 712-718).

The discussion of potentials should be accordingly improved.

 2. The authors refer to sites with visibly lower potentials as possibly being linked to sites of hydrogen trapping. That is a bit misleading and needs to be discussed better.

First of all, it is possible that also other heterogeneities on the surface and their different response on hydrogen and re-oxidation may play a role here.

But even when assuming a link to hydrogen activity, trapping beneath the surface is per se usually not expected to cause lower potentials at the surface. The simple reason is that trapped hydrogen is prevented to reach the surface.

If low potential sites are seen right from the beginning, they could rather indicate regions of faster permeation. But that would require that this is not only the case in the surface region, but also to a great extend in the bulk in that region. This seems rather unlikely.

Hence, the other possibility is traps with small binding energy (shallow traps).  If they are filled before the end of the charging then the small delay in H reaching the surface does not play a role anymore and the oxide is partially reduced in the same way as the rest of the surface. But after stopping charging, the relatively fast release form shallow traps provides a higher H rate still reaching the surface.  However, the contrast between these sites and the rest of the surface should change with time, as different decay behaviours are expected (see Koyama et al., J. Electrochem. Soc. 162(12) (2015) C638-C647; Krieger et al. Acta Mater. 144 (2018) 235-244).

 3. Lower potential at cracks: it is expected, seeing the duration of the low potential in the cracked area: indicates severe formation of traps also in bulk region, maintaining a long-term supply of H.

The statement : …”Zakroczymski [39] modelling of hydrogen state in steel showed that effusion from the traps takes significantly more time relatively interstitial hydrogen.” is hence only one part of the explanation. It is important that the amount of H desorbing form the traps is high enough to cause the low potentials., i.e. the amount of H trapped needs to be very high, i.e. the trap density needs to be very high.

Author Response

Dear Mr. Reviewer, thank you very much for kind review. The authors are agreeing with all your comments and the corresponding changes and discussions are added to the manuscript. No doubts your valuable work was useful for improving of the article.

This is certainly a work suitable for publication, but some issues need to be addressed.

 1.First of all, here Kelvin probe measurements have been performed on the passive layers of an HSS and a mild steel have been performed on the hydrogen charging side and the other side (referred to as permeation/diffusion side).

That the potential of the oxides changes due to the presence of sufficient activity of hydrogen is already well known. The main message seems to be here that the according effect changes with time after charging. This should maybe made clearer. 

The additional explanations were added.

Thus, we can suppose that hydrogen in the steel can influence on the potential due to change the oxide film thickness (Eq. 2) and the ratio of activities of iron species (Eq. 3). “ and “On the other hand, it is possible that the steel electrode potential can be controlled by the reactions of hydrogen with the oxidizers containing in air (moisture and oxygen) [9,16, 20].”

 Certainly, one can be a bit more precise. And hence the according discussion can be improved.

The part discussing the potential determing reaction was additionally re-written.

 This is especially true for the charging side. On the charging side the oxide is (during cathodic charging) most likely completely reduced (certainly the iron oxide part, for the HSS it depends on the oxide of the less noble elements what really remains during charging).  That should be better discussed.

This part was clarified. Thank you.

This is already the first difference between the two sides: on the charging side a completely new oxide forms in the moment the charging is stopped, but under the influence of initially very high hydrogen activity. On the diffusion side the existing oxide is partially reduced by the hydrogen.

At the same time, as correctly pointed out by the authors, oxygen (from the air) will cause (re-)oxidation (on both sides).

Yes, thank you. These points were clarified in the manuscript.

Concerning potential, it can be said is, that as long as the hydrogen permeation towards the surface is still relatively high, i.e. most likely in the first 1000 s where the lowest potentials are measured, the potentials are most certainly defined by the equilibrium between oxygen reduction and hydrogen oxidation, modified by the different catalytic properties of the surface. This is because due to high H permeation rates anodic H oxidation is possible as counter reaction, i.e. band bending will be negligible.

Yes, thank you, completely agree. The point was clarified.

Reviewer 2 Report

The authors have carried out Scanning Kelvin probe mapping of surface potential to visualize the locations of hydrogen in mild steel and martensitic HSS. They have explained their results reasonably.

Author Response

Dear Mr. Reviewer, thank you very much for kind review. The authors are agreeing with all your comments and the corresponding changes and discussions are added to the manuscript. No doubts your valuable work was useful for improving of the article.

On noble metals such as Pd such a formation of ORR-HOR equilibrium was recently reported (see Zhong et al., Chemelectrochem 8(4) (2021) 712-718).

The discussion of potentials should be accordingly improved.

The reference was addad to manuscript and discussed.

Author Response

Dear Mr. Reviewer, thank you very much for kind review. The authors are agreeing with all your comments and the corresponding changes and discussions are added to the manuscript. No doubts your valuable work was useful for improving of the article.

The authors refer to sites with visibly lower potentials as possibly being linked to sites of hydrogen trapping. That is a bit misleading and needs to be discussed better.

First of all, it is possible that also other heterogeneities on the surface and their different response on hydrogen and re-oxidation may play a role here.

Yes, it is right, preliminary to experiment the surfaces were scanned and non-uniformity (can be related to othert effects) were detected.  

But even when assuming a link to hydrogen activity, trapping beneath the surface is per se usually not expected to cause lower potentials at the surface. The simple reason is that trapped hydrogen is prevented to reach the surface.

Yes it is in steady state conditions at the particular temperature. Before equilibrium reached the local fluxes can appear.

If low potential sites are seen right from the beginning, they could rather indicate regions of faster permeation. But that would require that this is not only the case in the surface region, but also to a great extend in the bulk in that region. This seems rather unlikely.

Yes, this point was additionally discussed but different experiments show this possibility.

Hence, the other possibility is traps with small binding energy (shallow traps).  If they are filled before the end of the charging then the small delay in H reaching the surface does not play a role anymore and the oxide is partially reduced in the same way as the rest of the surface. But after stopping charging, the relatively fast release form shallow traps provides a higher H rate still reaching the surface.  However, the contrast between these sites and the rest of the surface should change with time, as different decay behaviours are expected (see Koyama et al., J. Electrochem. Soc. 162(12) (2015) C638-C647; Krieger et al. Acta Mater. 144 (2018) 235-244).

Thank you, the reference added to the article and the contrasts in the potential map was discussed in the manuscript.

  1. Lower potential at cracks: it is expected, seeing the duration of the low potential in the cracked area: indicates severe formation of traps also in bulk region, maintaining a long-term supply of H.

The statement : …”Zakroczymski [39] modelling of hydrogen state in steel showed that effusion from the traps takes significantly more time relatively interstitial hydrogen.” is hence only one part of the explanation. It is important that the amount of H desorbing form the traps is high enough to cause the low potentials., i.e. the amount of H trapped needs to be very high, i.e. the trap density needs to be very high.

Yes the additional phrases explaining the hydrogen effusion were added.